# Prognosis of the Ipsilesional Corticospinal Tracts with Preserved Integrities at the Early Stage of Cerebral Infarction: Follow Up Diffusion Tensor Tractography Study

**DOI:** 10.3390/healthcare10061096

**Published:** 2022-06-13

**Authors:** Sung Ho Jang, Hye Rin Seo, Dong Hyun Byun

**Affiliations:** 1Department of Physical Medicine and Rehabilitation, College of Medicine, Yeungnam University, Daemyungdong, Namku, Daegu 42415, Korea; strokerehab@hanmail.net; 2Sinchon Severance Hospital, Younsei University College of Medicine, Seoul 03722, Korea; hyerinseo17@naver.com

**Keywords:** diffusion tensor imaging, corticospinal tract, cerebral infarction, hemiparesis, prognosis

## Abstract

This study examined the prognosis of the ipsilesional corticospinal tracts (CSTs) with preserved integrities at the early stage of cerebral infarction using follow-up diffusion tensor tractography (DTT). Thirty-one patients with a supratentorial infarction were recruited. DTT, Motricity Index (MI), modified Brunnstrom classification (MBC), and functional ambulation category (FAC) were performed twice at the early and chronic stages. The patients were classified into two groups based on the integrity of the ipsilesional CST on the second DTT: Group A (24 patients; 77.4%)—preserved integrity and Group B (7 patients; 22.6%)—disrupted integrity. No significant differences in MI, MBC, and FAC were observed between groups A and B at the first and second evaluations, except for FAC at the first evaluation (*p* > 0.05). MI, MBC, and FAC at the second evaluation were significantly higher than at the first evaluation in both groups A and B (*p* < 0.05). On the second DTT, one patient (4.2%) in group A showed a false-positive result, whereas five patients (71.4%) in group B had false-negative results. Approximately 20% of patients showed disruption of the ipsilesional CST at the chronic stage. However, the clinical outcomes in hand and gait functions were generally good. Careful interpretation considering the somatotopy of the ipsilesional CST is needed because of the high false-negative results on DTT at the chronic stage.

## 1. Introduction

Stroke is a medical condition in which poor blood flow to the brain causes neuron death; there are two main types of stroke: ischemic due to lack of blood flow (87%), and hemorrhagic due to bleeding (13%) [1,2]. The main risk factors for stroke comprise hypertension, hypercholesterolemia, smoking, obesity, diabetes, and cardiac arrhythmia [2,3,4]. Each year, nearly 795,000 people experience a new or recurrent stroke in the United States (approximately 2.7%); approximately 610,000 of these are first attacks, and the other 185,000 people are recurrent attacks [2]. Stroke is a leading cause of adult disability, and previous studies have reported that motor deficits are common after stroke (82% of patients) and are linked with reduced quality of life [5,6,7]. The burden of stroke is increasing despite incredible progress and advancements in stroke management [8,9]. A forecast reported that stroke-related medical costs will exceed 183 billion US dollars annually by 2030 in the United States [9,10].

The motor weakness in stroke patients is caused mainly by an injury to the corticospinal tract (CST), which is the most critical neural tract for the motor function in the human brain [11]. Hence, clarification of the prognosis of the ipsilesional CST at the early stage of stroke is clinically important for predicting the prognosis of motor weakness [12]. Functional magnetic resonance imaging (MRI), which stimulates corticospinal neurons or interneurons synapsing on corticospinal neurons originating from the motor cortex, and transcranial magnetic stimulation have been commonly used to evaluate the CST state [13,14,15]. Functional MRI is limited because patients with severe weakness cannot perform the motor tasks required for cortical activation [13,14]. Transcranial magnetic stimulation also has poor spatial resolution and the possibility of false-negative results due to the excessively high threshold at the early stage of stroke [13,14].

The introduction of diffusion tensor tractography (DTT), which is derived from diffusion tensor imaging (DTI), has enabled three-dimensional (3D) reconstructions of the CST [16]. DTT methods for reconstructing the CST have been reported to have excellent repeatability and reproducibility [17]. As a result, many studies have reported the utility of DTT-based CST analysis in predicting the prognosis of motor weakness in stroke patients, using the DTT parameters and configurational integrity of the ipsilesional CST on DTT [6,18,19,20,21,22,23,24,25]. Configurational analysis of the CST allows easy and quick application compared to analyzing the DTT parameters in the clinical field. Hence, many DTT-based studies have shown that preserved integrity of the ipsilesional CST is an essential condition for the good outcome of motor weakness in stroke patients [22,23,24,25]. On the other hand, the DTT technique may overestimate or underestimate the ipsilesional CST because the CST is reconstructed with the setting conditions of the threshold value of fractional anisotropy and the trajectory angle for termination of tracking [25,26]. Furthermore, DTT for the CST can show false-positive results when DTT is performed before Wallerian degeneration at the early stages of stroke [27]. Many DTT-based studies have reported the longitudinal changes to the ipsilesional CST from the early to the chronic stages of stroke [6,18,19,20,21,28]. However, the majority of studies have reported the changes in the DTT parameters [6,18,19,20,21]. By contrast, little is known about the changes in the configurational integrity of the ipsilesional CST, which is easily applicable in clinics for stroke patients [28].

This study examined the prognosis of the ipsilesional CSTs with the clinical outcome, in which configurational integrities were preserved at the early stage of cerebral infarction, using follow-up DTTs.

## 2. Methods

### 2.1. Subjects

Thirty-one consecutive patients (24 males, 7 females; mean age, 56.54 ± 13.57 years; range, 21–75 years) with no history of neurologic/psychiatric disease and traumatic brain injury were enrolled in this study. The patients were recruited consecutively according to the following inclusion criteria: (1) first-ever stroke; (2) infarction confined to the supratentorial area, which involved the CST pathway (the primary sensorimotor cortex, centrum semiovale, corona radiata, and posterior limb of the internal capsule); (3) DTI was obtained during an early stage of cerebral infarction (less than 30 days after onset) and a chronic stage (more than 90 days after onset); (4) age at the time of onset, 20–75 years; (5) hemiparesis contralateral to the hemisphere in the cerebral infarction; (6) preserved integrity of the ipsilesional CST from the primary sensorimotor cortex to the medullary pyramid. Patients with a hemorrhagic transformation, and serious medical complications, such as pneumonia or cardiac problems from the onset to final evaluation, were excluded. We collected the clinical and DTI data of the patients from February 2004 to December 2021, at the rehabilitation center of Yeungnam University Hospital. This study was performed retrospectively in accordance with the requirements of the Declaration of Helsinki research guidelines, and the institutional review board of Yeungnam University Hospital approved the study protocol (YUMC 2021-03-014).

### 2.2. Clinical Evaluation

Clinical evaluations were conducted twice at the time of DTI scanning using the Motricity Index (MI) for the motor function of the ipsilesional extremities, the modified Brunnstrom classification (MBC) for motor function of the ipsilesional hand, and functional ambulation category (FAC) for the walking ability. The MI score (maximum 100) is converted from the Medical Research Council scoring system [29,30,31]. The total MI score is the average of the upper MI score (average of prehension, elbow flexor, and shoulder flexor) and lower MI score (ankle dorsiflexor, knee extensor, and hip flexor) [29]. The modified Brunnstrom classification (MBC) was used to categorize the function of the ipsilesional hand. The scores are as follows: 1, unable to move fingers voluntarily; 2, able to move the fingers voluntarily; 3, able to close the hand voluntarily, unable to open the hand; 4, able to grasp a card between the thumb and the medial side of the index finger, and able to extend the fingers slightly; 5, able to pick up and hold a glass, and able to extend fingers; 6, able to catch and throw a ball in a near-normal fashion, and able to button and unbutton a shirt [30]. The standardized functional ambulation category (FAC) was used to quantify the walking ability based on the level of assistance needed during a 15 min walk. The scores are as follows: 0, non-ambulatory; 1, needs continuous support from only one person; 2, needs intermittent support from one person; 3, needs only verbal supervision; 4, needs help on stairs and uneven surfaces; and 5, can walk independently anywhere [31]. The reliability and validity of the MI, MBC, and FAC are well established [29,30,31].

A good outcome was defined when a patient could execute the complete grasp-release movement of the affected hand (MBC; 5–6) and walk independently (FAC; 4–5) at the second clinical evaluation. By contrast, a poor outcome was defined when a patient could not execute a complete grasp-release movement of the affected hand (MBC; 1–4) and could not walk completely independently (FAC; 1–3).

### 2.3. Diffusion Tensor Imaging and Tractography

DTI data were acquired twice at the early (mean 12.58 ± 4.34 days) and chronic (mean: 287.87 ± 240.68 days) stages after the stroke onset using a six-channel head coil on a 1.5 T Philips Gyroscan Intera scanner (Philips, Best, The Netherlands) by single-shot echo-planar imaging. For each of the 32 non-collinear diffusion-sensitizing gradients, 60 contiguous slices were acquired parallel to the anterior commissure–posterior commissure line. The imaging parameters were as follows: acquisition matrix = 96 × 96, reconstructed to matrix = 192 × 192, field of view = 240 × 240 mm, repetition time = 10,398 ms, time to echo = 72 ms, parallel imaging reduction factor (SENSE factor) = 2, echo-planar imaging factor = 59, and b = 1000 s/mm^2^, number of excitations = 1, and thickness = 2.5 mm. Fiber tracking was performed using fiber assignment by continuous tracking, a three-dimensional fiber reconstruction algorithm within the Philips PRIDE software (Philips Medical Systems, Best, The Netherlands). For CST analysis, the seed ROI was placed on the CST portion of the pontomedullary junction, and the target ROI was positioned on the CST portion of the anterior midpons [17]. The termination criteria used for fiber tracking were a FA value of <0.2 and an angle of <60 degrees [16,17].

The patients were classified into two groups based on the ipsilesional CST findings on the second DTT; Group A—the ipsilesional CST originating from the primary sensorimotor cortex connected to the medullary pyramid through around the infarcted lesion and Group B—the CST was discontinued at or around the infarcted lesion, or reconnected to the other cerebral cortex except for the primary sensorimotor cortex (Figure 1A).

Regarding the compatibility between the configurational integrity of the ipsilesional CST on the second DTT and clinical outcome at the second clinical evaluation, a false-positive result was defined when a patient showed poor hand and gait functions, even though the integrity of the ipsilesional CST was preserved [32,33]. By contrast, a false-negative result was defined when a patient revealed good hand or gait functions, even though the integrity of the ipsilesional CST had been disrupted [32,33].

### 2.4. Statistical Analysis

Statistical analysis was performed using SPSS 21.0 for Windows (SPSS, Chicago, IL, USA). Considering the relatively small sample size, non-parametric statistical tests were used for the analysis. A chi-squared test (χ^2^; sex and age) was used to examine the demographic data. A Mann–Whitney test was performed to compare the age and clinical data (the MI, MBC, and FAC) at the first and second clinical evaluations. The Wilcoxon signed-rank test was used to determine the significance of changes in the MI, MBC, and FAC scores between the first and second clinical evaluations. Statistical significance was considered for *p* values < 0.05.

## 3. Results

On the second DTT, 24 patients (77.4%, 19 males, 5 females; mean age, 59.12 ± 11.89 years; range, 31–75 years) were classified as group A, and the remaining 7 patients (22.6%, 5 males, 2 females; mean age, 47.71 ± 16.16 years; range, 21–69 years) belonged to Group B (Table 1). No significant differences in age and sex were detected between groups A and B (*p* > 0.05).

Table 2 compares the clinical data (MI, MBC, and FAC scores) between the first and second clinical evaluations. No significant differences in the MI, MBC, and FAC scores were observed between groups A and B at the first and second clinical evaluations, except for the FAC score at the first clinical evaluation (*p* > 0.05). The FAC score was significantly higher in group A than in group B at the first clinical evaluation (*p* < 0.05). The MI, MBC, and FAC scores at the second clinical evaluation were significantly higher than those at the first clinical evaluation in groups A and B, respectively (*p* < 0.05).

Table 3 summarizes the clinical outcomes of the hand and gait functions at the second clinical evaluation. Among the 31 patients, 21 patients (67.7%) had good outcomes in hand function, and 24 patients (77.4%) had good outcomes in gait function. Seventeen patients (54.8%) showed good outcomes in both hand and gait functions (group A: 16 patients (51.6%); group B: 1 patient (3.3%)). Four patients (12%) revealed good outcomes only in the hand function (group A: 2 patients (6.4%); group B: 2 patients (6.4%)), whereas seven patients (22%) presented good outcomes only in gait function (group A: 5 patients (16.1%), group B; 2 patients (6.4%)).

On the second DTT, 1 patient (4.2%, poor outcome in both hand and gait functions) among 24 patients in group A showed a false positive result, whereas 5 out of 7 patients (71.4%: four patients, good outcome in hand or gait function; one patient, good outcomes in both hand and gait functions) in group B revealed false-negative results (Figure 1B,C).

## 4. Discussion

This study investigated the prognosis of the ipsilesional CSTs with preserved integrities at the early stages of cerebral infarction using follow-up DTTs and the clinical outcomes. The results are summarized as follows. (1) On the second DTT, among the 31 patients, 24 patients (77.4%) presented preserved integrity of the ipsilesional CST (group A), and the remaining 7 patients (22.6%) revealed a disrupted integrity of the ipsilesional CST (group B). (2) At the first and second clinical evaluations, no differences in the motor, hand, and gait functions were observed between groups A and B except for gait function (better in group A than group B) at the first clinical evaluation. (3) Motor, hand, and gait functions were improved at the second clinical evaluation more than at the first clinical evaluation in both groups, respectively. (4) At the second clinical evaluation, among the 31 patients, 21 patients (67.7%) presented good outcomes in hand function, and 24 patients (77.4%) revealed good outcomes in gait function. Regarding integrity preservation of the ipsilesional CST, in patients with preserved integrity (group A, 24 patients), 18 patients (75.0%) showed good outcomes in hand function, whereas 21 patients (87.5%) revealed good outcomes in gait function. By contrast, in patients with disrupted integrity (group B, seven patients), three patients (42.9%) had good outcomes in hand function, and three patients (42.9%) had good outcomes in gait function. (5) On the second DTT, 1 patient (4.2%) among 24 patients with preserved integrity (group A) showed a false-positive result, whereas 5 patients (71.4%) among 7 patients with disrupted integrity (group B) revealed false-negative results.

The patients recruited in this study showed generally good outcomes at the chronic stage; approximately 70% of patients presented good hand function, and approximately 80% showed good gait function. In approximately 20% of patients, the integrity of the ipsilesional CST was disrupted at the chronic stage. A comparison of the clinical outcomes regarding the integrity preservation of the ipsilesional CST at the chronic stage showed that approximately 80% of patients with preserved integrity revealed a good outcome in hand and gait functions. In contrast, approximately 40% of patients with disrupted integrity revealed good outcomes in hand and gait functions. On the other hand, significant differences were not detected between the two groups. These results might be affected by the small number of patients with disrupted integrity (7 patients) compared to patients with preserved integrity (21 patients).

Among the 24 patients who showed preserved integrity of the ipsilesional CST at the chronic stage (group A), 8 patients showed poor hand or gait functions at the second clinical evaluation (five patients, poor hand function; two patients, poor gait function; one patient, poor hand and gait functions). Seven out of eight patients who had a poor hand or gait function had an infarction corresponding to the hand or leg somatotopic area of the CST at the corona radiata, which corresponds to each poor function (Figure 1(C1)) [33]. Hence, a patient with a selective lesion on the hand or leg somatotopic areas of the CST can result in poor outcomes, which correspond to the related somatotopy of the CST, even though the integrity of the CST is preserved at the chronic stage [33]. On the other hand, one patient with spared integrity who had poor hand and gait functions at the chronic stage appeared to be due to the spared CST pathway through the large infarcted lesion at the subcortical white matter (Figure 1(B2)). Therefore, a patient spared only the CST pathway surrounded by an adjacent large infarcted lesion can show a false-positive result on DTT at the chronic stage. Among the seven patients who showed disrupted integrity of the ipsilesional CST on DTT at the chronic stage (group B), two patients (28.6%) who revealed poor outcomes in hand and gait functions had infarcted lesions involving the hand and leg somatotopic areas of the CST (Figure 1(B3)) [32,33]. Among the remaining five (71%) patients, four patients (57.1%) who showed good hand or gait function had an infarcted lesion corresponding to the leg or hand somatotopic areas of the CST at the corona radiata, respectively [32,33]. These results suggest that when a patient has a spared somatotopic area of the CST from the infarcted lesion, they have a good outcome of the corresponding function at the chronic stage, even if the integrity of the ipsilesional CST is disrupted at the chronic stage. By contrast, one patient with a good outcome in hand and gait functions at the chronic stage was attributed to a spared CST, even though the patient had a large infarcted lesion at the anterior and middle corona radiata (Figure 1(C4)) [32,33]. Consequently, five (71.4%) out of seven patients with disrupted integrity (group B) showed false-negative results at the chronic stage. As a result, DTT for the ipsilesional CST at the chronic stage of cerebral infarction showed a high false-negative rate with a low rate of false-positive results. The high false-negative results at the chronic stage appeared to be attributed to the characteristics of the DTT reconstruction method; the trajectory of DTT was reconstructed based on the given conditions in terms of fractional anisotropy and angle [25,26]. This study recruited patients with lesions involving the ipsilesional CST along the CST pathway. Therefore, DTT for the ipsilesional CST can be reconstructed through an infarcted lesion or peri-infarct edematous area at the early stages of the cerebral infarction (Figure 1 A-group B-green arrows). On the other hand, the integrity of the ipsilesional CST can be disrupted around the infarcted lesion after resolution of the peri-infarct edema and settlement of the infarcted lesion at the chronic stage, even though the ipsilesional CST is spared (Figure 1 A-group B-blue arrows). Hence, the combined use of transcranial magnetic stimulation would be necessary to compensate for this problem of DTT [25]. Further studies on this topic will be needed.

After introducing DTI, several DTT-based studies have reported the changes in the ipsilesional CST in stroke patients, using follow-up DTTs [6,18,19,20,21,28]. The majority of these studies have reported on the changes in the DTT parameters (particularly, fractional anisotropy (FA) value) of the ipsilesional CST from the acute or early to the chronic stage of stroke [6,18,19,20,21]. A few studies among the above studies have reported the decrease of FA value of the ipsilesional CST on follow-up DTTs, and the decrease of FA value was positively correlated with the motor deficit at the chronic stage of cerebral infarction [6,18,19,20]. In contrast, Ma et al. (2014) investigated the clinical usefulness of follow-up DTTs in predicting the motor outcome in patients with basal ganglia hemorrhage [21]. They found that the FA value of the ipsilesional CST at onset was significantly lower in the poor outcome group than in the good outcome group, and the FA value gradually decreased in the poor outcome group until 90 days after onset, while it continuously increased in the good outcome group. To the best of the authors’ knowledge, only one study reported the changes in the integrity of the ipsilesional CST from the early to the chronic stage in stroke patients [28]. Jung and Jang reported the longitudinal changes in the ipsilesional CST from the early to the chronic stage of intracerebral hemorrhage in 44 patients with severe hemiparesis [28]. DTTs for the ipsilesional CST were classified into three types: integrity of the ipsilesional CST was preserved, integrity of the ipsilesional CST was disrupted around the hematoma, and the ipsilesional CST showed Wallerian degeneration following integrity disruption. Among the 14 patients who showed preserved integrity of the ipsilesional CST, the integrity of the ipsilesional CST was preserved in 12 patients (85.7%) at the chronic stage, and the remaining 2 patients (14%) showed a disruption of the ipsilesional CST at the chronic stage. The change in the incidence of ipsilesional CST was similar to the present study, even though the pathological characteristics of cerebral infarction and intracerebral hemorrhage are different. On the other hand, detailed clinical and radiologic findings were not provided in this study. As a result, as far as we are aware, this is the first study to demonstrate the prognosis of the ipsilesional CST along with the clinical outcomes in patients with cerebral infarction. Nevertheless, the limitations of this study need to be considered. First, DTT analysis can overestimate or underestimate the neural fiber status in areas with crossing fibers or fiber complexity [26]. Second, the relatively small number of subjects, particularly those in group B, was too small compared to group A. Therefore, further studies involving a larger number of subjects are needed.

## 5. Conclusions

In conclusion, this study investigated the prognosis of ipsilesional CST with preserved integrities at the early stages of cerebral infarction along with the clinical outcomes. Approximately 20% of patients showed disruption of the ipsilesional CST at the chronic stage. The clinical outcomes were generally good in terms of hand and gait functions, but the patients with preserved integrity showed a better trend than the patients with a disrupted integrity in clinical outcome. Careful interpretation considering the relationship between the infarcted lesion and somatotopy of the ipsilesional CST is needed because the clinical outcomes in patients with a selective somatotopic lesion of the CST were closely related to the corresponding clinical outcome. Moreover, DTT at the chronic stage can show high false-negative results [32,33]. On the other hand, high individual variability of somatotopy of the CST should be considered [33].

## Figures and Tables

**Figure 1 healthcare-10-01096-f001:**
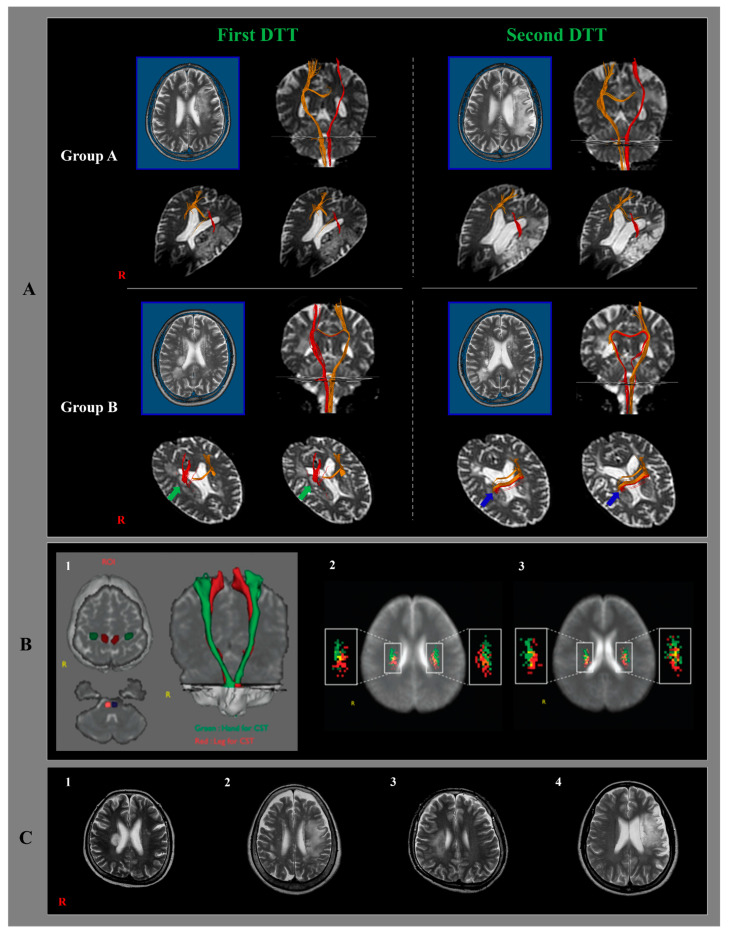
(**A**): Classification of diffusion tensor tractography (DTT) for the integrity of the ipsilesional corticospinal tract (CST) (red color) of the representative patients of each group. T2-weighted brain MR images: the left side of the upper row. Coronal images of DTT: the right side of the upper row, and axial images of DTT at or around the infarcted lesion level (the lower row). Group A: preserved integrity of the ipsilesional CST on the second DTT. Group B: disrupted integrity of the ipsilesional CST on the second DTT (green arrows: the ipsilesional CST descends through the infarcted lesion, blue arrows: the integrity of the ipsilesional CST is disrupted. (**B**): DTT shows locations for the upper and lower extremities of the CST, and measurement of somatotopic organization at the different corona radiata (CR) levels. (1) The seed regions of interests (ROIs) are given at the precentral knob (green) for the upper extremity and at the mediodorsal part (red) for the lower extremity. Target ROI is given at the anterior portion of the pontine, and the CSTs are reconstructed in both hemispheres (green: upper extremity for the CST, red: lower extremity for the CST). (2) CST probabilistic maps are shown at the upper CR level. (3) CST probabilistic maps are shown at the lower CR level (green: highest probabilistic locations for the upper extremity, red: highest probabilistic locations for the lower extremity) (reprinted with permission from [33]). (**C**): (1) Infarcted lesion in the hand somatotopic area of the CST. (2) some spared the CST pathway through the large infarcted lesion at the subcortical white matter. (3) Infarcted lesion involving the hand and leg somatotopic areas of the CST. (4) Large infarcted lesion at the anterior and middle corona radiata.

**Table 1 healthcare-10-01096-t001:** Demographic data in groups A and B.

		A	B	
No		24	7	-
Sex	(M/F)	19/5	5/2	*p >* 0.05
Age		59.12 (11.89)	47.71 (16.16)	*p >* 0.05
Infarct side	(Right/Left)	11/13	4/3	*p >* 0.05
Infarct location		24	7	*p >* 0.05
	MCA	7	3	
	Cortex	1	-	
	Centrum semiovale	1	-	
	Corona radiate	13	4	
	Posterior limb	2	-	

Values represent mean (±standard deviation); MCA: Middle Cerebral Artery.

**Table 2 healthcare-10-01096-t002:** Changes in clinical data in groups A and B.

	Group A		Group B					
	1st	2nd	1st	2nd				
MI	53.72 (22.83)	80.30 (13.93)	35.42 (25.76)	68.97 (16.15)	*p *>* 0.05* ^a^	*p* > 0.05 ^b^	*p* < 0.05 ^c^*	*p* < 0.05 ^d^*
MBC	2.95 (2.15)	4.83 (1.30)	2.10 (1.95)	3.71 (1.70)	*p* > 0.05 ^a^	*p* > 0.05 ^b^	*p* < 0.05 ^c^*	*p* < 0.05 ^d^*
FAC	1.45 (1.35)	4.20 (0.65)	0.21 (0.56)	3.71 (0.95)	*p* < 0.05 ^a^*	*p* > 0.05 ^b^	*p* < 0.05 ^c^*	*p* < 0.05 ^d^*

Values represent the mean (±standard deviation). MI: Morticity Index; MBC: modified Brunnstrom classification; FAC: functional ambulation category. Mann–Whitney test was used to compare 1st and 2nd evaluation in group A and group B: ^a^ 1st evaluation in group A and group B; ^b^ 2nd evaluation in group A and group B; ^c^ 1st and 2nd evaluation in group A; ^d^ 1st and 2nd evaluation in group B; *: Significant difference between two time points, *p* < 0.05.

**Table 3 healthcare-10-01096-t003:** Distribution of good outcomes at the second clinical evaluation.

	Total	Group A	Group B
Patient No.	31	24	7
	(100%)	(77.4%)	(22.6%)
Both	17	16	1
	(54.8%)	(51.6%)	(3.2%)
Hand	4	2	2
	(12.9%)	(6.4%)	(6.4%)
Gait	7	5	2
	(22.5%)	(16.1%)	(6.4%)

Values represent mean (±standard deviation).

## Data Availability

Data are available on request due to privacy/ethical restrictions.

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
