# Peer review of "Prognosis of the Ipsilesional Corticospinal Tracts with Preserved Integrities at the Early Stage of Cerebral Infarction: Follow Up Diffusion Tensor Tractography Study"

_healthcare, 2022, doi:10.3390/healthcare10061096_

Round 1
Reviewer 1 Report
The article I think could be interesting for your readers, but there are many changes to be made in the introduction and discussion. This is so the authors can redo it, but the article is to be removed from submission and done again. I will give you some recommendations that I hope will help the authors to improve it.
In the introduction it is necessary that they make a more concise explanation of what stroke is, types, incidence, risk factors and degree of disability that it generates, as well as talking about the socio-sanitary costs derived from its treatment and care. The introduction lacks current references and does not go deep enough into the pathology it is dealing with. Being a topic of great interest, there are many current bibliographical references, with less than 5 years that can be used and that your readers would appreciate. It would be interesting to make a brief definition of what transcranial magnetic stimulation is to make it easier to read.
Where are the exclusion criteria for this article specified?
It is not specified where the study was conducted or on what date.
At the level of the discussion, it would be advisable to redo it again because it is repetitive and does not provide the reader with any type of extra information. You may want to use the articles you use in reference 7-15 to compare your results and provide some clinical evidence.
I hope this information is of interest to you and can help improve the article.
Author Response
Point 1)
In the introduction it is necessary that they make a more concise explanation of what stroke is, types, incidence, risk factors and degree of disability that it generates, as well as talking about the socio-sanitary costs derived from its treatment and care. The introduction lacks current references and does not go deep enough into the pathology it is dealing with. Being a topic of great interest, there are many current bibliographical references, with less than 5 years that can be used and that your readers would appreciate. It would be interesting to make a brief definition of what transcranial magnetic stimulation is to make it easier to read.
Answer: We totally agree with the reviewer’s comment. So, we revised as follows.
- Introduction
Stroke is a medical condition in which poor blood flow to the brain causes neuron death, and there are two main types of stroke: ischemic due to lack of blood flow (87%), and hemorrhagic due to bleeding (13%) [1,2]. The main risk factors for stroke com-prise hypertension , hypercholesterolemia, smoking, obesity, diabetes, and cardiac ar-rhythmia [2-4]. Each year, nearly 795,000 people experience a new or recurrent stroke in the United States (approximately 2.7%); approximately 610,000 of these are first attacks, and the rest 185,000 people are recurrent attacks [2]. Stroke is a leading cause of adult disability, and previous studies reported that motor deficits are common after stroke (82% of patients) and are linked with reduced quality of life [5-7]. The burden of stroke is in-creasing despite incredible progress and advancements in stroke management [8,9]. A forecast reported that stroke-related medical costs will exceed 183 billion US dollars annually by 2030 in the United States [9-10].
The motor weakness in stroke patients is caused mainly by an injury to the cortico-spinal tract (CST), which is the most critical neural tract for the motor function in the human brain [11]. Hence, clarification of the prognosis of the ipsilesional CST at the early stage of stroke is clinically important for predicting the prognosis of motor weakness [12]. Functional magnetic resonance imaging (MRI) which stimulates corticospinal neurons or interneurons synapsing on corticospinal neurons originating from the motor cortex and transcranial magnetic stimulation have been commonly used to evaluate the CST state [13-15].
References
1 National Heart, Lung, and Blood Institute Homepage. Available online: https://web.archive.org/web/20150218230259/http:/www.nhlbi.nih.gov/health/health-topics/topics/stroke/ (archived from the original on 18 February 2015. retrieved 26 February 2015). 1 National Heart, Lung, and Blood Institute Homepage. Available online: https://web.archive.org/web/20150218230259/http:/www.nhlbi.nih.gov/health/health-topics/topics/stroke/ (archived from the original on 18 February 2015. retrieved 26 February 2015).
2 Tsao, C.W.; Aday, A.W.; Almarzooq, Z.I.; Alonso, A.; Beaton, A.Z.; Bittencourt, M.S.; Boehme, A.K.; Buxton, A.E.; Carson, A.P.; Commodore-Mensah, Y.; et al. Heart disease and stroke statistics-2022 update: a report from the american heart association. Circulation 2022, 145, e153-e639, doi: 10.1161/CIR.0000000000001052. Epub 2022 Jan 26.
3 Bogiatzi, C.; Hackam, D.G.; McLeod, A.I.; Spence, J.D. Secular trends in ischemic stroke subtypes and stroke risk factors. Stroke 2014, 45, 3208-3213, doi: 10.1161/STROKEAHA.114.006536. Epub 2014 Sep 11.
4 Anwer, S.; Waris, A.; Gilani, S.O.; Iqbal, J.; Shaikh, N.; Pujari, A.N.; Niazi, I.K. Rehabilitation of upper limb motor impairment in stroke: a narrative review on the prevalence, risk factors, and economic statistics of stroke and state of the art therapies. Healthcare (Basel) 2022, 10, 190, doi: 10.3390/healthcare10020190.
5 Rathore, S.; Hinn, A.; Cooper, L.; Tyroler, H.; Rosamond, W. Characterization of incident stroke signs and symptoms: findings from the atherosclerosis risk in communities study. Stroke 2002, 33, 2718–2721, doi: 10.1161/01.str.0000035286.87503.31.
6 Thomalla, G.; Glauche, V.; Koch, M.A.; Beaulieu, C.; Weiller, C.; Rother, J. Diffusion tensor imaging detects early wallerian degeneration of the pyramidal tract after ischemic stroke. Neuroimage 2004, 22, 1767-1774, doi: 10.1016/j.neuroimage.2004.03.041.
7 Saposnik, G.; Levin, M. Virtual reality in stroke rehabilitation: a meta-analysis and implications for clinicians. Stroke 2011, 42, 1380–1386, doi: 10.1161/STROKEAHA.110.605451.
8 Singh, R.J.; Chen. S.; Ganesh, A.; Hill, M.D. Long-term neurological, vascular, and mortality outcomes after stroke. Int J Stroke 2018, 13, 787–796, doi: 10.1177/1747493018798526.
9 Strilciuc, S.; Grad, D.A.; Radu, C.; Chira, D.; Stan, A.; Ungureanu, M.; Gheorghe, A.; Muresanu, F.D. The economic burden of stroke: a systematic review of cost of illness studies. J Med Life 2021, 14, 606-619, doi: 10.25122/jml-2021-0361.
10 Ovbiagele, B.; Goldstein, L.B.; Higashida, R.T.; Howard, V.J.; Johnston, S.C.; Khavjou, O.A.; Lackland, D.T.; Lichtmat, J.H.; Mohl, S.; Sacco, R.L; et al. Forecasting the future of stroke in the united states: a policy statement from the american heart association and american stroke association. Stroke 2013, 44, 2361– 2375, doi: 10.1161/STR.0b013e31829734f2.
11 Jang, S.H. The corticospinal tract from the viewpoint of brain rehabilitation. J Rehabil Med 2014, 46,193-199, doi: 10.2340/16501977-1782.
12 Duncan, P.W.; Goldstein, L.B.; Matchar, D.; Divine, G.W.; Feussner, J. Measurement of motor recovery after stroke. Outcome assessment and sample size requirements. Stroke 1992, 23,1084-1089, doi:10.1161/01.str.23.8.1084.
13 Jang, S.H. A review of motor recovery mechanisms in patients with stroke. NeuroRehabilitation 2007, 22, 253-259.
14 Macdonell, R.A.; Jackson, G.D.; Curatolo, J.M.; Abbott, D.F.; Berkovic, S.F.; Carey, L.M.; Syngeniotin, A.; Fabinyi, G.C.; Scheffer, I.E. Motor cortex localization using functional MRI and transcranial magnetic stimulation. Neurology 1999, 53, 1462-1467, doi: 10.1212/wnl.53.7.1462.
15 Diana, M.; Raij, T.; Melis, M.; Nummenmaa, A.; Leggio, L.; Bonci, A. Rehabilitating the addicted brain with transcranial magnetic stimulation. Nat Rev Neurosci 2017 18, 685-693, doi: 10.1038/nrn.2017.113.
Point 2)
Where are the exclusion criteria for this article specified?
Answer: Thank you for the reviewer’s comment. We already described in the methods as follows (underline).
- Methods
2.1. Subjects
Thirty-one consecutive patients (24 males, seven females; mean age, 56.54± 13.57 years; range, 21-75 years) with no history of neurologic/psychiatric disease and traumatic brain injury were enrolled in this study. The patients were recruited consecutively according to the following inclusion criteria: (1) first-ever stroke; (2) infarction confined to the supratentorial area, which involved the CST pathway (the primary sensorimotor cortex, centrum semiovale, corona radiata, and posterior limb of the internal capsule); (3) DTI was obtained during an early stage of cerebral infarction (less than 30 days after onset) and a chronic stage (more than 90 days after onset); (4) age at the time of onset, 20-75 years; (5) hemiparesis contralateral to the hemisphere in the cerebral infarction; (6) preserved integrity of the ipsilesional CST from the primary sensorimotor cortex to the medullary pyramid. Patients with a hemorrhagic transformation, and serious medical complications, such as pneumonia or cardiac problems from the onset to final evaluation, were excluded.
Point 3)
It is not specified where the study was conducted or on what date.
Answer: Thank you for the reviewer’s comment.
- Methods
2.1. Subjects
Thirty-one consecutive patients (24 males, seven females; mean age, 56.54± 13.57 years; range, 21-75 years) with no history of neurologic/psychiatric disease and traumatic brain injury were enrolled in this study. The patients were recruited consecutively ac-cording to the following inclusion criteria: (1) first-ever stroke; (2) infarction confined to the supratentorial area, which involved the CST pathway (the primary sensorimotor cortex, centrum semiovale, corona radiata, and posterior limb of the internal capsule); (3) DTI was obtained during an early stage of cerebral infarction (less than 30 days after onset) and a chronic stage (more than 90 days after onset); (4) age at the time of onset, 20-75 years; (5) hemiparesis contralateral to the hemisphere in the cerebral infarction; (6) preserved integrity of the ipsilesional CST from the primary sensorimotor cortex to the medullary pyramid. Patients with a hemorrhagic transformation, and serious medical complications, such as pneumonia or cardiac problems from the onset to final evaluation, were excluded. We collected the clinical and DTI data of the patients from february, 2004 to december, 2021 at the rehabilitation center of Yeungnam university hospital. This study was performed retrospectively in accordance with the requirements of the Declaration of Helsinki research guidelines, and the institutional review board of a Yeungnam university hospital approved the study protocol (YUMC 2021-03-014).
Point 4)
At the level of the discussion, it would be advisable to redo it again because it is repetitive and does not provide the reader with any type of extra information. You may want to use the articles you use in reference 7-15 to compare your results and provide some clinical evidence.
I hope this information is of interest to you and can help improve the article.
Answer: We totally agree with the reviewer’s comment. The reference 12~14 were not follow up DTT study. The reference 7~10 was to investigate the Wallerian degeneration and the reference 11 was to investigate the prediction of motor outcome using follow up DTTs. So, we revised as follows.
After introducing DTI, several DTT-based studies have reported the changes in the ipsilesional CST in stroke patients, using follow-up DTTs [18-22,29]. The majority of these studies have reported on the changes in the DTT parameters (particularly, frac-tional anisotropy[FA] value) of the ipsilesional CST from the acute or early to the chronic stage of stroke [18-22]. A few studies among the above studies have reported the decrease of FA value of the ipsilesional CST on follow up DTTs, and the decrease of FA value was positively correlated with the motor deficit at the chronic stage of cerebral in-farction [18-21]. In contrast, Ma et al.[2014] investigated the clinical usefulness of follow up DTTs in predicting the motor outcome in patients with basal ganglia hemorrhage [22]. They found that the FA value of the ipsilesional CST at onset was significantly lower in the poor outcome group than in the good outcome group, and the FA value gradually decreased in the poor outcome group until 90 days after onset, while it continuously in-creased in the good outcome group. To the best of the authors’ knowledge, only one study reported the changes in the integrity

Reviewer 2 Report
In this study, the authors investigated the prognosis of ipsilesional CST with preserved integrities at the early stages of cerebral infarction along with the clinical outcomes.
The title is too long.
P-values are not exact and only 0.05 is presented as a cut-off.
The sample size is not large enough. There are only 7 patients in group B.
Author Response
Point 1)
The title is too long.
Answer: We totally agree with the reviewer’s comment. We revised as follows.
Prognosis of the ipsilesional corticospinal tracts with preserved integrities at the early stage of cerebral infarction: follow up diffusion tensor tractography study
Point 2)
P-values are not exact and only 0.05 is presented as a cut-off.
Answer: We totally agree with the reviewer’s comment. We revised as follows.
|
Table 1. Demographic data in groups A and B |
||||
|
A |
B |
|||
|
No |
|
24 |
7 |
- |
|
Sex |
(M / F) |
19 / 5 |
5 / 2 |
p>0.05 |
|
Age |
|
59.12 (11.89) |
47.71 (16.16) |
p>0.05 |
|
Infarct side |
(Right / Left) |
11 / 13 |
4 / 3 |
p>0.05 |
|
Infarct location |
|
24 |
7 |
p>0.05 |
|
|
MCA |
7 |
3 |
|
|
|
Cortex |
1 |
- |
|
|
|
Centrum semiovale |
1 |
- |
|
|
|
Corona radiate |
13 |
4 |
|
|
|
Posterior limb |
2 |
- |
|
Table 2. Changes of clinical data in groups A and B.
|
|
Group A |
Group B |
||||||
|
|
1st |
2nd |
1st |
2nd |
|
|||
|
MI |
53.72 |
80.30 |
35.42 (25.76) |
68.97 |
p>0.05a |
p>0.05b |
p<0.05c* |
p<0.05d* |
|
MBC |
2.95 |
4.83 (1.30) |
2.10 (1.95) |
3.71 |
p>0.05a |
p>0.05b |
p<0.05c* |
p<0.05d* |
|
FAC |
1.45 (1.35) |
4.20 (0.65) |
0.21 (0.56) |
3.71 |
p<0.05a* |
p>0.05b |
p<0.05c* |
p<0.05d* |
Point 3)
The sample size is not large enough. There are only 7 patients in group B.
Answer: We totally agree with the reviewer’s comment. We also thought this is a limitation of this study. However, the comparison between two groups was not main purpose of this study. The main purpose was to investigate the prognosis of the ipsilesional CSTs in which configurational integrities were preserved at the early stage of cerebral infarction which can see frequently in clinics for stroke patients. In addition, scanning of follow up DTIs with clinical data is not easy in clinics. Actually, we collected these data for approximately 18 years from February 2004, to December 2021 although we usually manage over 150 new stroke patients annually. So, we already described this as a limitation in the discussion as follows (underline).
- Discussion
Nevertheless, the limitations of this study need to be considered. First, DTT analysis can overestimate or underestimate the neural fiber status in areas with crossing fibers or fiber complexity [27]. Second, the relatively small number of subjects, particularly those in group B, was too small compared to group A. Therefore, further studies involving a larger number of subjects are needed.

Reviewer 3 Report
The study entitled "Prognosis of the ipsilesional corticospinal tracts in which the configurational integrities are preserved at the early stage of cerebral infarction: follow up diffusion tensor tractography study" is interesting and represents a clear demonstration of the current limits of diffusion tensor tractography.
The main limit is the low number of patients but it remains difficult to perform several DTI for patients. This limitation clearly appeared in the discussion section.
Another limit is the application of only one DTT reconstruction algorithm: if the authors can perform another DTT reconstruction using a different software, it would strengthen the conclusions of this study.
The figures should be improved: the presentation of the whole series (especially considering the low number of included patients) will be a nice addition, if possible on a normalized template to ease the understanding.
The last sentence has to mitigate the usefulness of DTT in the context of stroke rehabilitation: in front of this study, it seems impossible to adequately predict motor function from DTT...
I recommend a revision of this manuscript to reach publication standard.
Author Response
Point 1)
The main limit is the low number of patients but it remains difficult to perform several DTI for patients. This limitation clearly appeared in the discussion section.
Answer: Thank you for your encouraging comment. We also thought this is a limitation of this study. Scanning of follow up DTIs with clinical data is not easy in clinics. Actually, we collected these data for approximately 18 years from February, 2004, to December, 2021 although we usually manage over 150 new stroke patients annually.
Point 2)
Another limit is the application of only one DTT reconstruction algorithm: if the authors can perform another DTT reconstruction using a different software, it would strengthen the conclusions of this study.
Answer: We totally agree with the reviewer’s comment. Using DTI, our lab has researched for approximately past 20 years. We have mainly used three DTT reconstruction programs (Pride, DTI studio, and FSL). Among these programs, we have experienced “Pride” which we used in this study and implemented in Philips MR machine, is most appropriate for the reconstruction of the corticospinal tract in terms of clinical correlation and neuroanatomy. Furthermore, the optimalities of ROI locations and reconstruction conditions (the threshold value of FA and the trajectory angle for termination of tracking) for reconstruction of the corticospinal tract which we used in this study, have been well demonstrated by our previous studies and other researchers [16]. The excellent reliability was also demonstrated by previous studies [17]. Although we totally agree with the reviewer’s comment, the main purpose of this study was not compare the DTT analysis programs.The main purpose was to investigate the prognosis of the ipsilesional CSTs in which configurational integrities were preserved at the early stage of cerebral infarction which can see frequently in clinics for stroke patients. So, we are obliged to turn down additional analysis using other programs. However, if you recommend again, we would like to analysis using other programs at next revision.
16 Kunimatsu, A.; Aoki, S.; Masutani, Y.; Abe, O.; Hayashi, N.; Mori, H.; Masumoto, T.; Ohtomo, K. The optimal trackability threshold of fractional anisotropy for diffusion tensor tractography of the corticospinal tract. Magn Reson Med Sci 2004, 3, 11-17, doi: 10.2463/mrms.3.11.
17 Seo, J.P.; Kwon, Y.H.; Jang, S.H. Mini-review of studies reporting the repeatability and reproducibility of diffusion tensor imaging. Investig Magn Reson Imaging 2019, 23, 1125199. doi: https://doi.org/10.13104/imri.2019.23.1.26
Point 3)
The figures should be improved: the presentation of the whole series (especially considering the low number of included patients) will be a nice addition, if possible on a normalized template to ease the understanding.
Answer: Thank you for the reviewer’s comment. We think the presentation of whole series appears not appropriate because the number of total DTT is 62 series. On the other hand, using normalized template also appears not appropriate because our patients had various lesion locations of cerebral infarction (the primary sensorimotor cortex, centrum semiovale, corona radiata, and posterior limb of the internal capsule). On figure 1-A, we think the confirmation of the integrity of the CST is necessary on the sagittal image and of the relation of the CST with lesion on the axial images. So, we modified figure 1-A a little without modification of the sagittal and axial images. Regarding figure 1-B, we could not modify the figure 1-B and it’s legand because this figure was reprinted from other published paper with permission of the journal (NeuroReport): we promised we will not modify the figure and its legand when we got permission. With regard to figure 1-C, it is necessary for discussion. However, if you recommend again the improvement of figure 1, we would like to modify the figure 1 at next revision.
Point 4)
The last sentence has to mitigate the usefulness of DTT in the context of stroke rehabilitation: in front of this study, it seems impossible to adequately predict motor function from DTT...
Answer: We totally agree with the reviewer’s comment. So, we revised as follows.
- Conclusions
In conclusion, this study investigated the prognosis of ipsilesional CST with preserved integrities at the early stages of cerebral infarction along with the clinical outcomes. Approximately 20 % of patients showed disruption of the ipsilesional CST at the chronic stage. The clinical outcomes were generally good in terms of hand and gait functions, but the patients with preserved integrity showed a better trend than the patients with a disrupted integrity in clinical outcome. Careful interpretation considering the relationship between the infarcted lesion and somatotopy of the ipsilesional CST is needed because the clinical outcomes in patients with a selective somatotopic lesion of the CST were closely related to the corresponding clinical outcome. Moreover, DTT at the chronic stage can show high false negative results [33,34]. On the other hand, high individual variability of somatotopy of the CST should be considered [34].
Please see the attachment

Reviewer 4 Report
The paper is well written and in general clear and scientifically sound, the authors investigate the outcome of post-stroke patients with intact CST. And they focus on the two different populations with preserved and not preserved CST at T1.
I think that bibliography on background literature should be more thorough
Additionally, DTI was used only to create 2 groups of patients, is it possible to search for a correlation between CST in DTI and clinical scores.
Finally, the groups are unbalanced in number and this could hinder results.
Author Response
Point 1)
I think that bibliography on background literature should be more thorough
Answer: We totally agree with the reviewer’s comment. We added more references as follows.
Stroke is a medical condition in which poor blood flow to the brain causes neuron death, and there are two main types of stroke: ischemic due to lack of blood flow (87%), and hemorrhagic due to bleeding (13%) [1,2]. The main risk factors for stroke com-prise hypertension , hypercholesterolemia, smoking, obesity, diabetes, and cardiac ar-rhythmia [2-4]. Each year, nearly 795,000 people experience a new or recurrent stroke in the United States (approximately 2.7%); approximately 610,000 of these are first attacks, and the rest 185,000 people are recurrent attacks [2]. Stroke is a leading cause of adult disability, and previous studies reported that motor deficits are common after stroke (82% of patients) and are linked with reduced quality of life [5-7]. The burden of stroke is in-creasing despite incredible progress and advancements in stroke management [8,9]. A forecast reported that stroke-related medical costs will exceed 183 billion US dollars annually by 2030 in the United States [9-10].
The motor weakness in stroke patients is caused mainly by an injury to the cortico-spinal tract (CST), which is the most critical neural tract for the motor function in the human brain [11]. Hence, clarification of the prognosis of the ipsilesional CST at the early stage of stroke is clinically important for predicting the prognosis of motor weakness [11]. Functional magnetic resonance imaging (MRI) which stimulates corticospinal neurons or interneurons synapsing on corticospinal neurons originating from the motor cortex and transcranial magnetic stimulation have been commonly used to evaluate the CST state [13-15].
References
1 National Heart, Lung, and Blood Institute Homepage. Available online: https://web.archive.org/web/20150218230259/http:/www.nhlbi.nih.gov/health/health-topics/topics/stroke/ (archived from the original on 18 February 2015. retrieved 26 February 2015). 1 National Heart, Lung, and Blood Institute Homepage. Available online: https://web.archive.org/web/20150218230259/http:/www.nhlbi.nih.gov/health/health-topics/topics/stroke/ (archived from the original on 18 February 2015. retrieved 26 February 2015).
2 Tsao, C.W.; Aday, A.W.; Almarzooq, Z.I.; Alonso, A.; Beaton, A.Z.; Bittencourt, M.S.; Boehme, A.K.; Buxton, A.E.; Carson, A.P.; Commodore-Mensah, Y.; et al. Heart disease and stroke statistics-2022 update: a report from the american heart association. Circulation 2022, 145, e153-e639, doi: 10.1161/CIR.0000000000001052. Epub 2022 Jan 26.
3 Bogiatzi, C.; Hackam, D.G.; McLeod, A.I.; Spence, J.D. Secular trends in ischemic stroke subtypes and stroke risk factors. Stroke 2014, 45, 3208-3213, doi: 10.1161/STROKEAHA.114.006536. Epub 2014 Sep 11.
4 Anwer, S.; Waris, A.; Gilani, S.O.; Iqbal, J.; Shaikh, N.; Pujari, A.N.; Niazi, I.K. Rehabilitation of upper limb motor impairment in stroke: a narrative review on the prevalence, risk factors, and economic statistics of stroke and state of the art therapies. Healthcare (Basel) 2022, 10, 190, doi: 10.3390/healthcare10020190.
5 Rathore, S.; Hinn, A.; Cooper, L.; Tyroler, H.; Rosamond, W. Characterization of incident stroke signs and symptoms: findings from the atherosclerosis risk in communities study. Stroke 2002, 33, 2718–2721, doi: 10.1161/01.str.0000035286.87503.31.
6 Thomalla, G.; Glauche, V.; Koch, M.A.; Beaulieu, C.; Weiller, C.; Rother, J. Diffusion tensor imaging detects early wallerian degeneration of the pyramidal tract after ischemic stroke. Neuroimage 2004, 22, 1767-1774, doi: 10.1016/j.neuroimage.2004.03.041.
7 Saposnik, G.; Levin, M. Virtual reality in stroke rehabilitation: a meta-analysis and implications for clinicians. Stroke 2011, 42, 1380–1386, doi: 10.1161/STROKEAHA.110.605451.
8 Singh, R.J.; Chen. S.; Ganesh, A.; Hill, M.D. Long-term neurological, vascular, and mortality outcomes after stroke. Int J Stroke 2018, 13, 787–796, doi: 10.1177/1747493018798526.
9 Strilciuc, S.; Grad, D.A.; Radu, C.; Chira, D.; Stan, A.; Ungureanu, M.; Gheorghe, A.; Muresanu, F.D. The economic burden of stroke: a systematic review of cost of illness studies. J Med Life 2021, 14, 606-619, doi: 10.25122/jml-2021-0361.
10 Ovbiagele, B.; Goldstein, L.B.; Higashida, R.T.; Howard, V.J.; Johnston, S.C.; Khavjou, O.A.; Lackland, D.T.; Lichtmat, J.H.; Mohl, S.; Sacco, R.L; et al. Forecasting the future of stroke in the united states: a policy statement from the american heart association and american stroke association. Stroke 2013, 44, 2361– 2375, doi: 10.1161/STR.0b013e31829734f2.
11 Jang, S.H. The corticospinal tract from the viewpoint of brain rehabilitation. J Rehabil Med 2014, 46,193-199, doi: 10.2340/16501977-1782.
12 Duncan, P.W.; Goldstein, L.B.; Matchar, D.; Divine, G.W.; Feussner, J. Measurement of motor recovery after stroke. Outcome assessment and sample size requirements. Stroke 1992, 23,1084-1089, doi:10.1161/01.str.23.8.1084.
13 Jang, S.H. A review of motor recovery mechanisms in patients with stroke. NeuroRehabilitation 2007, 22, 253-259.
14 Macdonell, R.A.; Jackson, G.D.; Curatolo, J.M.; Abbott, D.F.; Berkovic, S.F.; Carey, L.M.; Syngeniotin, A.; Fabinyi, G.C.; Scheffer, I.E. Motor cortex localization using functional MRI and transcranial magnetic stimulation. Neurology 1999, 53, 1462-1467, doi: 10.1212/wnl.53.7.1462.
15 Diana, M.; Raij, T.; Melis, M.; Nummenmaa, A.; Leggio, L.; Bonci, A. Rehabilitating the addicted brain with transcranial magnetic stimulation. Nat Rev Neurosci 2017 18, 685-693, doi: 10.1038/nrn.2017.113.
Point 2)
Additionally, DTI was used only to create 2 groups of patients, is it possible to search for a correlation between CST in DTI and clinical scores.
Answer: We totally agree with the reviewer’s comment. We anlysed and described the results in the below. We did not analysis for each group because subject number of group B was too small compared to group A. However, we are reluctant to add these correlation results because the main purpose of this study was to investigate the prognosis of the ipsilesional CSTs in which configurational integrities were preserved at the early stage of cerebral infarction which can see frequently in clinics for stroke patients. We think that these data is not compatible with the purpose of our study. Conversely, adding these data might make our manuscript more complicated without definite benefits for readers. However, if you recommend again, we would like to add these correlation results at next revision.
*Correlations between DTT parameters (FA, ADC, and TV) and clinical scores (MI, MBC, and FAC) at second evaluation: the results of the correlation between FA and total MI score showed moderate positive correlation (r = 0.494; p < 0.05). Also, moderate positive correlation was observed with MBC score (r = 0.479; p < 0.05). In correlation between TV of CST and clinical scores, moderate positive correlation was observed with total MI and MBC score (r = 0.464; p < 0.05; r = 0.424; p < 0.05). (FA: fractional anisotropy, ADC: apparent diffusion coefficient, TV: tract volume, MBC: modified Brunnstrom classification, FAC: functional ambulation category, MI: Motricity Index) [A correlation coefficient (r value) was interpreted as strong when > 0.50, as moderate when between 0.30 and 0.49, and weak when between 0.10 and 0.29: Reference 1: Cohen, Jacob. Statistical power analysis for the behavioral sciences. Routledge, 2013.
Point 3)
Finally, the groups are unbalanced in number and this could hinder results.
Answer: We totally agree with the reviewer’s comment. We also thought this is a limitation of this study. However, the comparison between two groups was not main purpose of this study. The main purpose was to investigate the prognosis of the ipsilesional CSTs in which configurational integrities were preserved at the early stage of cerebral infarction which can see frequently in clinics for stroke patients. In addition, scanning of follow up DTIs with clinical data is not easy in clinics. Actually, we collected these data for approximately 18 years from February, 2004, to December, 2021 although we usually manage over 150 new stroke patients annually. So, we already described this as a limitation in the discussion as follows (underline).
- Discussion
Nevertheless, the limitations of this study need to be considered. First, DTT analysis can overestimate or underestimate the neural fiber status in areas with crossing fibers or fiber complexity [27]. Second, the relatively small number of subjects, particularly those in group B, was too small compared to group A. Therefore, further studies involving a larger number of subjects are needed.
Please see the attachment

Reviewer 5 Report
In this manuscript, the authors focused on investigating the prognostic value of CSTs with preserved integrities at the early stage of cerebral infarction via usage follow-up DTT, an advanced MRI technique, and indicated that this study might be useful for prognosis prediction. Please see below for more comments:
- The sample size (n=31) is rather small even for clinical study.
- Elaboration is needed for the medical history of participants. Do they have any other pre-existing health conditions that might affect the accuracy of results in this study?
- The quality of Figure 1B (1) need to be improved. The text in this panel is eligible.
Minor comments:
- Please use punctuation correctly, an example can be found while reading the sentences in Abstract, “On the second DTT, one patient(4.2%) in group A showed a false-positive result, whereas five patients(71.4%) in group B had false-negative results.” should be “On the second DTT, one patient (4.2%) in group A showed a false-positive result, whereas five patients (71.4%) in group B had false-negative results”.
Author Response
The sample size (n=31) is rather small even for clinical study.
Answer: We totally agree with the reviewer’s comment. We also thought this is a limitation of this study. However, the comparison between two groups was not main purpose of this study. The main purpose was to investigate the prognosis of the ipsilesional CSTs in which configurational integrities were preserved at the early stage of cerebral infarction which can see frequently in clinics for stroke patients. In addition, scanning of follow up DTIs with clinical data is not easy in clinics. Actually, we collected these data for approximately 18 years from February, 2004, to December, 2021 although we usually manage over 150 new stroke patients annually. So, we already described this as a limitation in the discussion as follows (underline).
- Discussion
Nevertheless, the limitations of this study need to be considered. First, DTT analysis can overestimate or underestimate the neural fiber status in areas with crossing fibers or fiber complexity [27]. Second, the relatively small number of subjects, particularly those in group B, was too small compared to group A. Therefore, further studies involving a larger number of subjects are needed.
Point 2)
Elaboration is needed for the medical history of participants. Do they have any other pre-existing health conditions that might affect the accuracy of results in this study?
Answer: We totally agree with the reviewer’s comment. We already described and revised as follows.
- Methods
2.1. Subjects
Thirty-one consecutive patients (24 males, seven females; mean age, 56.54± 13.57 years; range, 21-75 years) with no history of neurologic/psychiatric disease and traumatic brain injury were enrolled in this study.
Point 3)
The quality of Figure 1B (1) need to be improved. The text in this panel is eligible.
Answer: We totally agree with the reviewer’s comment. However, we could not modify the figure 1-B and it’s legand because this figure was reprinted from other published paper with permission of the journal (NeuroReport).
Point 4)
Please use punctuation correctly, an example can be found while reading the sentences in Abstract, “On the second DTT, one patient(4.2%) in group A showed a false-positive result, whereas five patients(71.4%) in group B had false-negative results.” should be “On the second DTT, one patient (4.2%) in group A showed a false-positive result, whereas five patients (71.4%) in group B had false-negative results”.
Answer: We are sorry for our mistakes. So, we corrected our mistakes through the whole manuscript.
Please see the attachment

Round 2
Reviewer 1 Report
The authors have done a better job that gives their manuscript greater scientific rigor
Author Response
Thank you for your advice.
Reviewer 3 Report
As already suggested, the authors should provide a more detailed figure 1. Normalization is possible even in front of brain lesion. It would be interesting to have a spatial frequency map of the stroke localization.
The use of a second algorithm, if not possible on the whole series only on few patients, could help to overcome the low number of patients. If not possible, the authors have to add patients to the group B.
In short, I think that the provided revision does not take into account my comments. I thank in advance the authors who will make substantial changes in the manuscript.
Author Response
Point 1)
As already suggested, the authors should provide a more detailed figure 1. Normalization is possible even in front of brain lesion. It would be interesting to have a spatial frequency map of the stroke localization.
Answer: We totally agree with the reviewer’s comment. We normalized the initial and follow-up T2-weighted image data of 17 of 31 patients (group A: 12, group B: 5) to the normalized template using MRIcroGL. We could not get the above data from 13 of 31 patients because the data of these patients were erased from our hospital data (due to too old data). All patients (yellow), group A (red), and group B (light green) were marked as follow (below figure). However, we think this figure appears not appropriate for this manuscript because our patients had various lesion locations of cerebral infarction (the primary sensorimotor cortex, centrum semiovale, corona radiata, and posterior limb of the internal capsule). So, we are reluctant to insert this figure into figure 1. However, if you recommend again, we would like to insert this figure into figure 1 at next revision.
(we attached normalization figure in attachment)
Point 2)
The use of a second algorithm, if not possible on the whole series only on few patients, could help to overcome the low number of patients. If not possible, the authors have to add patients to the group B.
Answer: We totally agree with the reviewer’s comment. We also thought this is an important limitation of this study. Actually, we collected these data for approximately 18 years from February, 2004, to December, 2021 although we usually manage over 150 new stroke patients annually. Although we collected the data for approximately 18 years, the main reason of low number of subjects is that the patients who have preserved integrity of the ipsilesional CST when the patients start rehabiliation at the early stage of stroke at our rehabilitation center are very rare. On the other hand, majority of our stroke patients usually show discontiunation of the ipsi-lesional CST at the early of stroke. Because we recruited nine patient of group B for approximately 18 years, adding one patient of group B requires two years. So, adding more patients in group B is actually impossible at our rehabilitation center. So, we would like to ask you consider our situation.
"Please see the attachment."

Reviewer 5 Report
The authors have addressed all of my concerns.
Author Response
Thank you for your advice.